# Comparison of the performance of four clinical prediction rules for mortality in patients with COVID-19

Johan Azañero-Haro[1,2*☉], Alonso Soto[1,2☉]

**1** Instituto de Investigación en Ciencias Biomédicas, Facultad de medicina, Universidad Ricardo Palma, Lima, Perú, **2** Departamento de medicina interna, Hospital Nacional Hipolito Unanue, Lima, Perú

☉ These authors contributed equally to this work.
* johan.azanero@urp.edu.pe

## Abstract

### Background

Clinical prediction rules integrate clinical and laboratory variables to estimate outcomes, facilitating decision-making and optimizing resources, especially in high-demand settings. We aimed to validate and compare the performance of four mortality prediction scores -ISARIC-4C, CALL, SEIMC, q-CSI- in a Peruvian cohort of unvaccinated hospitalized COVID-19 pneumonia patients during the initial pandemic wave.

### Methods

We performed a retrospective cohort study based on a secondary analysis of data from a previous study (March-December 2020). To ensure a robust and standardized head-to-head comparison, we utilized a complete-case analysis (n = 1,074). Selection bias was rigorously assessed by comparing the analytic sample with excluded patients. Each score's performance was evaluated using sensitivity, specificity, predictive values, likelihood ratios, area under the receiver operating characteristic curve (AUROC), and robust calibration metrics, including the calibration intercept (α) and the calibration slope (β). The ISARIC-4C score was used as an international reference standard for benchmarking.

### Results

Among 3,074 hospitalized patients, 1,074 had complete data for all four scores; no clinically significant differences were found between this group and excluded participants, indicating a representative sample. The cohort was mainly male (67.9%) with a median age of 59 years. The q-CSI score showed the best discrimination (AUROC 0.85, 95% CI: 0.83–0.87), significantly better than ISARIC-4C (0.82, 95% CI: 0.80–0.85), SEIMC (0.78, 95% CI: 0.75–0.81), and CALL (0.69, 95% CI: 0.66–0.72) (p < 0.0001). Direct comparison favored q-CSI over ISARIC-4C (p = 0.0016).

provided the original author and source are credited.

**Data availability statement:** All relevant data are within the paper and its Supporting information files.

**Funding:** The author(s) received no specific funding for this work.

While the Hosmer-Lemeshow (HL) test indicated a lack of fit for some models, robust calibration analysis confirmed accurate scaling (β for all scores was approx. 1.0) and adequate centering (α values with $p > 0.05$), supporting their clinical reliability.

## Conclusions

In this Peruvian cohort, the q-CSI score exhibited the best predictive performance and highest feasibility for in-hospital mortality among patients with COVID-19 pneumonia. While the HL test indicated a lack of fit, the analysis of the calibration α and β confirmed that the models are globally well-calibrated, supporting their utility for risk stratification. However, local adjustment is still necessary prior to clinical use in our setting. These findings provide a valuable baseline for resource optimization in resource-limited setting during pandemic waves.

## Introduction

Since the onset of the COVID-19 pandemic, healthcare systems worldwide have faced unprecedented challenges – particularly in low- and middle-income countries. In Latin America, Peru reported one of the highest mortality rates after Mexico, Brazil, and Bolivia [1]. The overwhelming burden of disease and death associated with SARS-CoV-2 highlighted the urgent need for effective tools to stratify risk early and optimize resource allocation [2]. During the most critical phases of the pandemic, a significant proportion of patients developed severe illness, often presenting with pneumonia and respiratory failure, requiring hospitalization and intensive care unit (ICU) admission. In such a context, timely and accurate mortality prediction became a crucial component for guiding clinical decisions and improving patient outcomes [3].

Clinical prediction rules play a key role in assessing the risk of adverse outcomes in various clinical settings. Scores such as APACHE II and SOFA are frequently used in intensive care, while tools like PSI, CURB-65, SMART-COP, and A-DROP have been applied to community-acquired pneumonia [4]. However, the unique clinical and pathophysiological features of COVID-19 showed the limitations of traditional scoring systems, prompting the development of new, more tailored models that incorporate a broader range of clinical and laboratory parameters [5,6]. Despite the availability of several promising predictive models, many rely on data that are not consistently available, such as specific biomarkers or imaging studies. This scarcity is particularly acute in low-resource environments, which restricts their routine use in settings with limited resources. Consequently, several studies have focused on evaluating simpler, more accessible scoring systems that maintain predictive accuracy while being practical for widespread application [7]. We emphasize that the validation of simple scores is paramount for improving triage in healthcare systems constrained by logistics and data completeness.

Among the proposed COVID-19-specific scores, some have gained traction due to their validation across diverse healthcare contexts. The CALL score [8] includes comorbidities, age, lymphocyte count, and lactate dehydrogenase levels. The

ISARIC-4C model [9] is a widely validated and internationally recognized reference standard often utilized in global clinical guidelines; it incorporates age, sex, comorbidities, respiratory rate, oxygen saturation, level of consciousness, urea, and C-reactive protein. The SEIMC score [10], developed by the Spanish Society of Infectious Diseases and Clinical Microbiology, includes age, oxygen saturation, neutrophil-to-lymphocyte ratio, estimated glomerular filtration rate, and comorbidity burden. Lastly, the q-CSI [11] is a rapid, respiratory-focused tool that uses respiratory rate, oxygen saturation, and oxygen flow rate. Though these models have demonstrated utility, their performance may vary according to demographic, epidemiological, and clinical factors specific to different populations. Therefore, external validation is an essential step before implementing their use in specific clinical settings, especially in regions with healthcare dynamics different from those where the models were originally developed [5,12]. Validation in Latin American cohorts is especially critical, given the historical underrepresentation of this region in predictive score literature. Such validation efforts not only enhance decision-making during public health emergencies but also support the adaptation of clinical strategies to specific settings, ultimately contributing to more effective, evidence-based care [6,12].

Furthermore, while clinical practice has evolved due to vaccination and new variants, data from these initial, immunologically naive (unvaccinated) populations remain vital for understanding the baseline pathogenesis of the disease and for informing preparedness efforts against future novel pathogens or emerging waves where population immunity might be low.

Our study aimed to evaluate and compare the performance of the aforementioned clinical scoring systems -CALL, ISARIC-4C, SEIMC, and q-CSI-for predicting 30-day in-hospital mortality among the initial wave of unvaccinated patients hospitalized with COVID-19 pneumonia at a referral hospital in Lima, Peru.

## Materials and methods

### Study subjects

We conducted a retrospective cohort study through a secondary analysis of data from a previously conducted and published study [7]. The original study was based on the review of medical records from patients hospitalized with a diagnosis of COVID-19 at the Hospital Nacional Hipolito Unanue (HNHU) between March and December 2020. The study population included patients aged ≥18 years with a confirmed diagnosis of COVID-19-associated pneumonia, established by Reverse Transcription Polymerase Chain Reaction (RT-PCR), serologic testing, or compatible clinical and radiological findings.

To ensure a rigorous head-to-head comparison between all models, patients with incomplete clinical or laboratory data required for the calculation of any of the four evaluated scores were excluded. This complete-case analysis resulted in a final comparative cohort of n = 1,074 patients. Multiple imputation was not performed due to the nature of the missingness in specialized laboratory markers, such as C-reactive protein (CRP) and lactate dehydrogenase (LDH), where the high proportion of missing data in specific variables was deemed too extensive to yield statistically robust estimates, prioritizing the analysis of directly observed data. A detailed summary of the proportion missing for each key predictor across the four scoring systems is provided in S1 Table, prioritizing the analysis of directly observed data for clinical reliability.

The required sample size was estimated assuming a minimum difference of 10% in sensitivity and specificity among the scores which was predefined as the minimum clinically relevant threshold that would justify prioritizing one score over another for clinical triage and risk stratification. This calculation was performed with a 95% confidence level and 80% statistical power, yielding a minimum of 582 patients (291 deceased and 291 survivors). However, all patients with complete data for the calculation of the four scores were included in the final analysis, ensuring adequate statistical power for the head-to-head comparison.

### Data collection and quality control

The study was conducted in the COVID-19-designated inpatient areas of the hospital. Data collection was coordinated with the hospital's medical records department. Physical medical charts were manually reviewed from May 16 to June 30,

2022. Information was recorded using a standardized digital form and subsequently transferred to an Excel spreadsheet for analysis.

Data abstraction was performed by the same core team of physician-investigators who possessed direct clinical experience in the COVID-19 hospitalization units. This contextual knowledge and internal standardization were crucial for the consistent and accurate interpretation of clinical and laboratory variables, particularly those obtained from handwritten physical records. Rigorous quality control was achieved through a 100% double data entry process, where two team members independently transcribed the information into separate electronic files. Subsequently, a systematic cross-verification was executed to detect all potential discrepancies between the two independent entries. Any differences identified were immediately reviewed by a senior investigator and resolved by consulting the original patient medical chart, culminating in a single, final validated dataset.

The study involved only retrospective review of medical records; therefore, informed consent was not obtained due to the retrospective nature of the study. One of the authors (JAH) had access to identifiable clinical records during data collection; however, all data were anonymized before inclusion in the final database and for subsequent analysis. The study protocol of the original investigation was approved by the Ethics Committee of the Hospital Nacional Hipolito Unanue (Letter No. 096-2021-CIEI-HNHU), which also covers the present secondary analysis.

## Measurements

The primary outcome for all scores was 30-day in-hospital mortality. For each patient, we calculated the CALL, ISARIC-4C, SEIMC, and q-CSI scores. Additionally, we collected sociodemographic characteristics, comorbidities, clinical presentation at admission, vital signs, and laboratory results obtained within the first 24 hours of hospitalization. A standardized data collection form was used to record patient information, including age, sex, symptom duration, comorbid conditions, clinical status at admission, hospital length of stay, initial laboratory findings, and final outcome (discharge, ICU admission, or death).

## Statistical analysis

Continuous variables were summarized as mean ± standard deviation or median and interquartile range (IQR), depending on their distribution, which was assessed using the Shapiro–Wilk or Kolmogorov–Smirnov test. Categorical variables were reported as absolute and relative frequencies.

To address potential selection bias, comparisons of baseline characteristics (including age, sex, and key clinical markers) were performed between the final analytic sample (n = 1,074) and those excluded due to missing data (n = 889). These results are presented in S2 Table. Comparisons between survivors and non-survivors (and between included and excluded subcohorts) were performed using Student's t-test or Mann–Whitney U test for continuous variables, and the chi-square test or Fisher's exact test for categorical variables.

The predictive performance of each score was evaluated using the binary outcome of mortality. To ensure a methodologically robust, head-to-head comparison of all four models, all subsequent analyses (discrimination and calibration) were performed exclusively on the complete-case cohort (n = 1,074) where data for all four scores were available. Discriminatory ability was measured using the area under the receiver operating characteristic curve (AUROC) with 95% confidence intervals(CI) [13]. The optimal cutoff point for dichotomization was calculated using the Youden index [14] to facilitate a standardized binary clinical decision ("high risk" vs. "low risk"). A direct comparison of each score's AUROC was performed against the ISARIC-4C score, as this is a widely validated and commonly used model for COVID-19 risk stratification, to determine the relative efficacy of the different scoring systems. Sensitivity, specificity, positive and negative predictive values, and positive and negative likelihood ratios were calculated based on the optimal cutoff point.

The calibration of each score was assessed using a tiered approach, prioritizing graphical and quantitative metrics over the traditional goodness-of-fit test. Decile-based calibration plots were used to visually compare predicted versus

observed mortality probabilities across the risk spectrum, Spearman correlation coefficients between predicted and observed mortality probabilities, and the Hosmer–Lemeshow (HL) test was calculated for primary reporting [15]. The calibration intercept (α) and the calibration slope (β) were calculated for each score using logistic regression of the outcome on the predicted log-odds (logit transformation) of the score [16]. All statistical analyses were performed using STATA version 18 [17].

## Results

A total of 3,074 patient medical records were assessed during the study period. After applying clinical exclusion criteria, 2,377 patients were identified. An additional 414 records were excluded due to early death/discharge, hospital withdrawal, or incomplete baseline clinical documentation, resulting in a source database of 1,963 patients. The feasibility for score calculation varied significantly across models due to the differential availability of clinical and laboratory parameters (See S1 Table). The q-CSI showed the highest data completeness (93.9%), while the ISARIC-4C and CALL scores were more frequently restricted by missing laboratory markers (71.7% and 72.3%, respectively). Consequently, the subcohorts for each model were: q-CSI (n = 1,844), SEIMC (n = 1,792), CALL (n = 1,420), and ISARIC-4C (n = 1,408). The final analytic cohort for the head-to-head comparison consisted of 1,074 complete cases, including 577 survivors and 497 deceased (Fig 1).

The baseline characteristics of the overall cohort have been previously published [7]. Briefly, the population was predominantly male (67.9%), with a median age of 59 years, and a high prevalence of type 2 diabetes mellitus, hypertension, and obesity. The most common symptoms were dyspnea, cough, and general malaise, while the most frequent physical examination findings included abnormal respiratory patterns and fever. Laboratory tests commonly revealed leukocytosis, lymphopenia, hyperglycemia, hyperferritinemia, and elevated inflammatory markers. The median length of hospital stay was 7 days, and the overall mortality rate was 46.3%.

Baseline characteristics (including age, sex, and key clinical markers) were compared between the final analytic cohort (n = 1,074) and those excluded due to missing data (n = 889). Excluded patients were slightly younger and had lower prevalence of diabetes than included patients (See S2 Table for the full comparison).

### Predictive performance of each score

Table 1 presents the predictive performance of four mortality scores (q-CSI, ISARIC-4C, SEIMC, and CALL) in the cohort of hospitalized patients with COVID-19 pneumonia. For each score, the Area Under the Receiver Operating Characteristic Curve (AUROC) is reported along with its 95% confidence interval. Additionally, performance metrics -including sensitivity, specificity, predictive values, and likelihood ratios- are reported for the optimal dichotomization cutoff identified through Youden index analysis. To maximize statistical power, these metrics were calculated using the maximal available number of patients in the subcohorts of each specific score.

### Comparison of the ROC curves of mortality prediction scores

In the group of 1,074 patients with complete data for the calculation of the four mortality prediction scores (q-CSI, ISARIC-4C, SEIMC, and CALL), a direct comparison of their discriminative ability was performed using ROC curve analysis (Fig 2). The AUROC for each score in this population were: 0.85 (95% CI: 0.83–0.87) for the q-CSI score, 0.82 (95% CI: 0.80–0.85) for the ISARIC-4C score, 0.78 (95% CI: 0.75–0.81) for the SEIMC score, and 0.69 (95% CI: 0.66–0.72) for the CALL score. The comparative analysis of the areas under the ROC curves revealed significant differences in the ability of the scores to discriminate between patients with and without mortality (p < 0.0001).

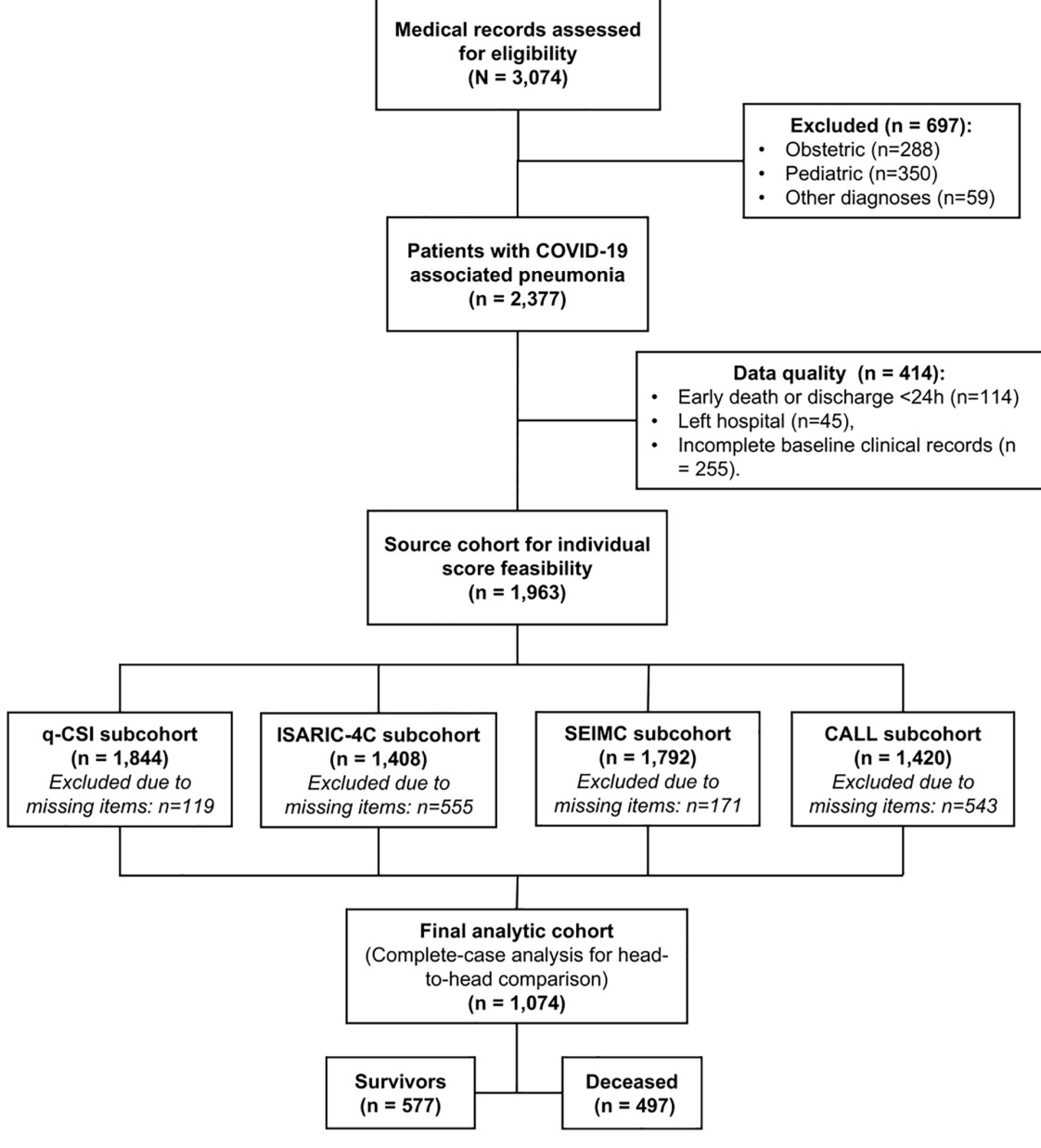

**Fig 1. Flowchart of study participant selection.** The diagram illustrates the recruitment and exclusion process of patients hospitalized with COVID-19 pneumonia at the Hospital Nacional Hipolito Unanue (Lima, Peru) between March and December 2020. The final analytic cohort (n = 1,074) represents the complete-case sample, including only participants with no missing values for any of the four clinical prediction rules evaluated, allowing for a robust head-to-head performance comparison.

## Comparison of predictive performance against the ISARIC-4C score

The predictive performance of the q-CSI, SEIMC, and CALL scores was directly benchmarked against the ISARIC-4C score, a widely validated international reference standard. Pairwise AUROC comparisons were conducted using the maximal available subcohorts for each model.

**Table 1. Performance of four mortality prediction rules in hospitalized patients with COVID-19 pneumonia. Maximal available subcohorts per score.**

| Score | AUROC (95% CI) | Optimal cutoff point* | Se (95% CI) | Sp (95% CI) | PPV (95% CI) | NPV (95% CI) | LR + (95% CI) | LR – (95% CI) |
|---|---|---|---|---|---|---|---|---|
| **q-CSI** (n = 1,844) | 0.86 (0.84–0.87) | ≥6 | 86.3 (83.8–88.4) | 70.9 (67.9–73.8) | 72.8 (69.9–75.5) | 85.1 (82.5–87.5) | 3.0 (2.7–3.3) | 0.19 (0.16–0.23) |
| **ISARIC-4C** (n = 1,408) | 0.81 (0.78–0.83) | ≥9 | 78.7 (75.5–81.8) | 69.1 (65.6–72.4) | 70.2 (66.8–73.5) | 77.8 (74.4–81.0) | 2.5 (2.3–2.9) | 0.31 (0.26–0.36) |
| **SEIMC** (n = 1,792) | 0.76 (0.74–0.78) | ≥8 | 64.2 (60.9–67.4) | 76.0 (73.2–78.7) | 70.7 (67.3–73.9) | 70.2 (67.3–73.0) | 2.7 (2.4–3.0) | 0.47 (0.43–0.52) |
| **CALL** (n = 1,420) | 0.68 (0.65–0.70) | ≥11 | 51.3 (47.4–55.1) | 79.1 (76.0–82.0) | 68.6 (64.3–72.6) | 64.6 (61.4–67.7) | 2.5 (2.1–2.9) | 0.62 (0.57–0.67) |

AUROC: Area under the receiver operating characteristic curve, Se: Sensitivity, Sp: Specificity, PPV: Positive predictive value, NPV: Negative predictive value, LR +: Positive likelihood ratio, LR-: Negative likelihood ratio. * Based on Youden Index.

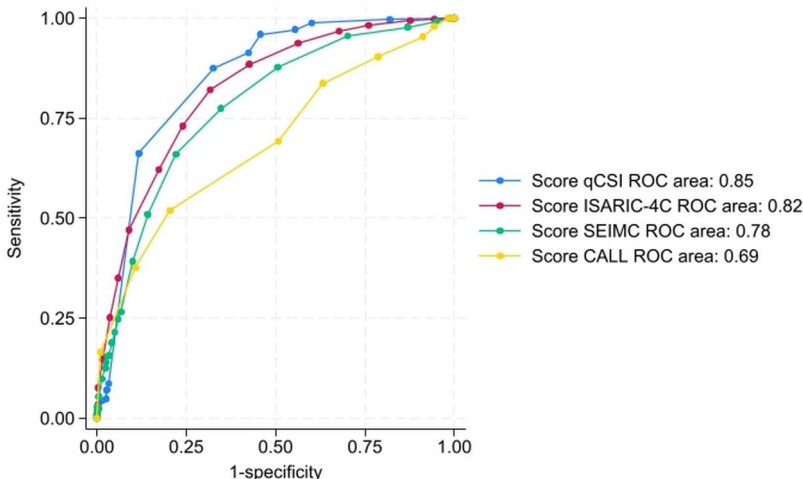

**Fig 2. Comparison of the ROC curves of the q-CSI, ISARIC-4C, SEIMC, and CALL scores in the subgroup of patients with complete data (n = 1074). Hospital Nacional Hipolito Unanue. March to December 2020. Lima, Peru.**

In the comparison between the q-CSI and ISARIC-4C scores (n = 1,408), the AUROC was 0.85 (95% CI: 0.83–0.87) and 0.81 (95% CI: 0.78–0.83), respectively, showing a statistically significant difference in favor of q-CSI (p = 0.002). When comparing the SEIMC score with the ISARIC-4C score (n = 1374), the AUROC was 0.76 (95% CI: 0.74–0.79) and 0.81 (95% CI: 0.78–0.83), respectively, with significantly superior performance for the ISARIC-4C score (p < 0.0001). Finally, the comparison between the CALL score and the ISARIC-4C score (n = 1101) showed an AUROC of 0.68 (95% CI: 0.65–0.71) and 0.82 (95% CI: 0.80–0.84), respectively, significantly favoring the ISARIC-4C score (p < 0.0001).

## Evaluation of the calibration of mortality prediction scores

The calibration of the four mortality prediction scores was evaluated using calibration plots (Fig 3) and Spearman rank correlations between the expected and observed mortality probabilities in each risk decile. Model fit was assessed using the Hosmer-Lemeshow (HL) goodness-of-fit test. Furthermore, recognizing the limitations of the HL test, we calculated and report the Calibration Intercept (α) and Calibration Slope (β) for each score, as suggested as more robust metrics used to

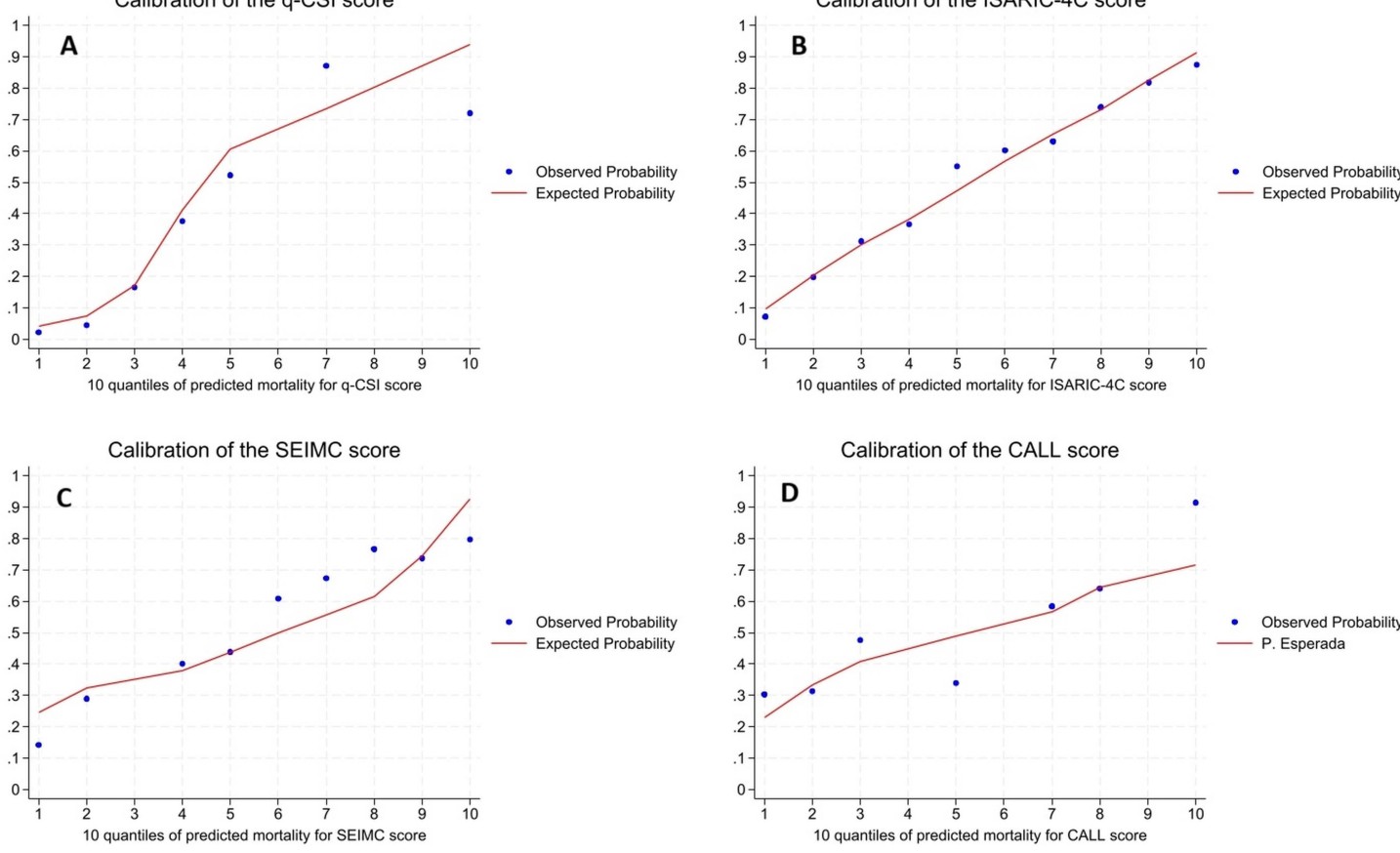

**Fig 3. Calibration plots for the q-CSI, ISARIC-4C, SEIMC, and CALL scores.** Expected and observed mortality probabilities by deciles are presented for each score's values.

assess potential systematic deviations and quantify the need for model recalibration in our setting, as recommended in the methodology literature.

Table 2 presents the summary of discrimination (AUROC) and the robust calibration metrics (α and β) for all four scores in the complete case cohort (n = 1,074).

For the q-CSI score (Fig 3A), the calibration plot showed a tendency to overestimate risk in the lower deciles and underestimate it in the upper deciles. The Spearman correlation between predicted and observed probabilities was high (r = 0.96; p = 0.003). However, the HL test revealed a statistically significant lack of fit (p < 0.001). The calibration intercept was −0.14 (95% CI: −0.29, 0.01) and the calibration slope was 1.02 (95% CI: 0.89, 1.16). These values confirm that the global bias (α) was not statistically significant (p = 0.074), and the risk scaling (β) was nearly ideal, challenging the overall implication of the HL test.

The ISARIC-4C score (Fig 3B) showed an overall adequate calibration pattern, with slight overestimation of the risk in the lower probability deciles. The Spearman correlation was excellent (r = 1.00; p < 0.001), and the HL test showed no lack of fit (p = 0.45), suggesting good calibration in the evaluated cohort. The calibration intercept was −0.13 (95% CI: −0.27, 0.01) and the calibration slope was 1.08 (95% CI: 0.94, 1.21), confirming near ideal values (Table 2)

For the SEIMC score (Fig 3C), the model tended to overpredict risk in the lower deciles, whereas predictions in the higher deciles were more variable and less accurate. The Spearman correlation was high (r = 0.98; p < 0.001), but the HL

**Table 2. Discrimination (AUROC) and calibration metrics (Intercept α and Slope β) of q-CSI, ISARIC-4C, SEIMC, and CALL scores in the complete-case cohort (n = 1,074).**

| Score | AUROC (95% CI) | HL Test p-value | Calibration intercept (α) | 95% CI (α) | P-value (α = 0) | Calibration slope (β) | 95% CI (β) |
|---|---|---|---|---|---|---|---|
| **q-CSI** (n=1,074) | 0.85 (0.83–0.87) | <0.001 | −0.14 | −0.29, 0.01 | 0.074 | 1.02 | 0.89–1.16 |
| **ISARIC–4C** (n = 1,074) | 0.82 (0.80–0.85) | 0.450 | −0.13 | −0.27, 0.01 | 0.078 | 1.08 | 0.94–1.21 |
| **SEIMC** (n=1,074) | 0.78 (0.75–0.81) | <0.001 | −0.04 | −0.17, 0.10 | 0.595 | 1.09 | 0.91–1.27 |
| **CALL** (n=1,074) | 0.69 (0.66–0.72) | <0.001 | −0.03 | −0.15, 0.10 | 0.690 | 1.07 | 0.86–1.28 |

AUROC: Area under the receiver operating characteristic curve. HL: Hosmer-Lemeshow.

statistic revealed a significant lack of fit (p < 0.001). However, the calibration intercept (α = −0.04; 95% CI: −0.17, 0.10) and the calibration slope (β = 1.09; 95% CI: 0.91, 1.27) confirmed that the global bias was statistically insignificant (p = 0.595) and the scale of risk was appropriate (Table 2).

Finally, the CALL score (Fig 3D) exhibited a pattern similar to q-CSI, with overestimation of the risk in the lower deciles and underestimation in the upper ones, along with greater variability in the observed rates per decile. The Spearman correlation was high (r = 0.96; p = 0.003), but a significant lack of fit was also observed according to the HL test (p < 0.001). The robust metrics showed minimal global deviation: calibration intercept (α = −0.03; 95% CI: −0.15, 0.10) and calibration slope (β = 1.07; 95% CI: 0.86, 1.28), indicating that the model is globally well-calibrated (Table 2).

## Discussion

Our study assessed the performance of the q-CSI, ISARIC-4C, SEIMC, and CALL scores for predicting 30-day mortality in patients hospitalized with COVID-19 pneumonia. The main finding was that the q-CSI score demonstrated the best discriminatory ability in our cohort, significantly outperforming the ISARIC-4C score, an internationally recognized reference standard. By utilizing a head-to-head comparison within a complete-case analytic cohort (n = 1,074), we ensured that the observed differences in performance were not artifactual results of varying denominators, but rather reflected the intrinsic predictive value of each model in our population. In addition to its good performance, the q-CSI score stands out for its simplicity and applicability in clinical practice, as it is based solely on three easily assessable parameters at admission: respiratory rate, oxygen saturation, and supplemental oxygen requirement. This simplicity contrasts with the complexity of ISARIC-4C, which requires multiple laboratory data that are often subject to high rates of missing data in crisis settings. As shown in S1 Table, the feasibility of calculating the ISARIC-4C was only 71.7% in our population, largely due to the limited availability of C-reactive protein (n = 319 missing). In contrast, the q-CSI had the best percentage of complete data (93.9%), underscoring its better feasibility for rapid triage in resource-constrained environments where laboratory bottlenecks are common.

The superior performance of the q-CSI in our Peruvian cohort of unvaccinated patients (AUROC 0.85 [0.83–0.87]) should be contrasted with its evaluation in other settings. In the q-CSI development study [11], an AUROC of 0.90 (0.85–0.96) was reported for predicting mortality. However, external validations in unvaccinated patients showed variability in its diagnostic performance. A North American cohort reported an AUROC of 0.71 [18], a European study found an AUROC of 0.75 [19], while a Latin American population showed an AUROC of 0.74 [20]. The differences in predictive performance between our study, the original development study, and other external validations may be explained by several factors [11,18–21]. First, the baseline characteristics of the cohorts may vary significantly in terms of age, comorbidities, and severity at hospital admission [22,23]. Second, the therapeutic protocols implemented at different centers may have influenced patient outcomes, and therefore, the scores' ability to predict mortality. Additionally, the prevalence of different SARS-CoV-2 variants may have varied geographically, which could have affected the accuracy of the prognostic scores

[24]. Finally, it is important to note that external validation of prognostic models often show lower results than those reported in the development cohort.

The direct comparison with ISARIC-4C, a widely validated and internationally recognized reference standard often utilized in global clinical guidelines, showed a significantly lower discriminatory capacity than q-CSI in our cohort. Our findings suggest that while ISARIC-4C remains a robust and high-quality tool, the simpler parameters of the q-CSI may be more closely aligned with the specific clinical phenotype and the operational reality of our setting. This contrasts with the findings of Covino et al. [19], where ISARIC-4C showed the best performance with an AUROC of 0.80 (similar to what we found in our study), while q-CSI achieved an AUROC of 0.75. Covino et al. focused on patients over 60 years old, which could partly explain the divergence in results. These discrepancies underscore the importance of external validation studies to assess the performance of scores in different populations [25]. It is crucial to thoroughly describe the baseline characteristics of each cohort, such as age, sex, comorbidities, and disease severity, to better understand differences in outcomes and facilitate the proper generalization of findings. Of note, our findings are related to an unvaccinated cohort from the initial pandemic waves. The predictive validity of these scores is dynamic, not static, and must be re-evaluated against the context of high population immunity, new variants, and the standardization of treatments (e.g., widespread corticosteroid use), which substantially alters the natural history and risk profile of the disease and consequently the performance of clinical scoring systems.

## Calibration and clinical applicability

Although discrimination, i.e., the ability of the scores to distinguish between high and low-risk patients, was acceptable in our cohort, calibration -which assesses the concordance between predicted probabilities and observed mortality rates- [15] showed significant variability among the evaluated models. A key methodological finding relates to the assessment of calibration beyond the Hosmer-Lemeshow (HL) test, which indicated a statistically significant lack of fit for the q-CSI, SEIMC, and CALL scores (p<0.001). We interpreted these results cautiously, as the HL test is highly sensitive to large sample sizes like our complete case cohort (n=1,074). Therefore, we also calculated the calibration intercept (α) and the calibration slope (β) to provide a more comprehensive assessment of systematic error [16].

Our result (Table 2) revealed that the global calibration of most scores was strong. Specifically, the β value for all four scores was found to be statistically indistinguishable from the ideal value of 1.0 (all 95% CIs included 1.0). Furthermore, the α values for the SEIMC and CALL scores were close to zero (e.g., CALL α=−0.03), indicating negligible global bias. While the q-CSI (α=−0.14) and ISARIC-4C (α=−0.13) showed a slight tendency toward overestimation, this deviation was not statistically significant (p>0.05).

The lack of fit observed in the HL test and the visual plots (Fig 3), such as the tendency for q-CSI and CALL to overestimate risk in the lower deciles, underscores the critical need for local intercept adjustment before these scores are implemented in populations distinct from their derivation cohorts. This observed lack of calibration implies that the scores may overestimate or underestimate the actual mortality risk in certain probability ranges, which requires local adjustment before widespread use [26].

Of note, the ISARIC-4C score was the only one that showed globally adequate calibration based on the HL test (p=0.45) as well as calibration intercept and slopes, with a very high correlation between predicted and observed probabilities. Therefore, even when a model demonstrates good discriminative performance, minor local inadequacy in calibration may limit its clinical applicability [3,27].

## Interpretation of cutoffs and clinical utility

Beyond discrimination and calibration, it is crucial to analyze other performance metrics to understand the clinical utility of clinical prediction rules. In our study, at their respective optimal cutoff-points, the q-CSI showed a sensitivity of 86.3% and a specificity of 70.9%, the ISARIC-4C showed a sensitivity of 78.7% and a specificity of 69.1%, the SEIMC showed

a sensitivity of 64.2% and a specificity of 76%, and the CALL showed a sensitivity of 51.3% and a specificity of 79.1%. Sensitivity, which indicates the ability of the score to correctly identify patients who will develop the adverse event (in this case, mortality), and specificity, which indicates the ability of the score to correctly identify patients who will not develop the adverse event (in this case, survival), are essential for evaluating predictive performance [28,29].

One aspect that deserves attention is that the scores evaluated were designed as risk stratification tools, proposing ordinal categories rather than a single dichotomous cutoff point. Our decision to derive new binary cutoffs using the Youden index was purely methodological and pragmatic, facilitating a standardized and objective head-to-head comparison of the predictive capacity of the four tools for the primary binary outcome in our cohort. While these categories can aid clinical decision-making, our analysis showed that, when used as binary thresholds to differentiate between patients at higher and lower risk of death, the predefined categories often failed to reach sensitivity levels considered clinically useful.

Our results reveal significant differences in the performance of the scores. The q-CSI and ISARIC-4C demonstrated higher sensitivity, while SEIMC and CALL scores showed higher specificity. The balance between sensitivity and specificity is key. Scores with higher sensitivity are valuable when it is essential to minimize false negatives (i.e., not overlooking patients who will die), making them effective as screening tools to 'rule out' the risk of mortality. Conversely, scores with high specificity are preferred when it is important to minimize false positives, making them effective as confirmatory tools to "rule in" the risk of mortality and identify high-risk patients for prioritized intervention [30].

The choice of score depends on the clinical context and the relative costs of false positives and negatives. For example, in a resource-limited setting, a score with high sensitivity might be favored in order to treat low-risk patients in less complex clinical environments. On the other hand, for decisions regarding admission to critical care units in situations with very limited availability, scores with high specificity may be helpful to prioritize those patients at higher risk of death. Given the epidemiological and therapeutic dynamics of COVID-19, periodic re-validation and recalibration of these scores are required to maintain their clinical utility over time.

## Limitations and strengths

Our study has limitations inherent to its retrospective and single-center design. Retrospective cohort studies are susceptible to selection and information biases, and being conducted in a single center may limit the generalizability of the findings to other populations or healthcare settings. Another limitation is the high percentage of excluded patients due to incomplete documentation. While this introduces a potential risk of selection bias, we rigorously compared baseline characteristics between the final analytic cohort (n = 1,074) and those excluded due to missing data (n = 889), as detailed in S2 Table. The patients included in the analysis were slightly older and had a higher prevalence of diabetes (27.9% vs. 22.7%, p = 0.008); however, these differences are clinically modest and likely reflect the clinical reality that patients with more severe disease or comorbidities underwent more comprehensive laboratory testing, thus having more complete records. Most importantly, since all four scores were evaluated on the exact same cohort, the comparative hierarchy of their performance remains internally valid for a head-to-head assessment.

A limitation regarding generalizability is that the cohort included exclusively unvaccinated patients and was conducted during the initial waves of the pandemic (pre-Delta, pre-Omicron). This makes it difficult to extrapolate the results to vaccinated populations, where clinical presentation, therapeutic response, and outcomes may have changed substantially due to population immunity, viral variants, and evolving standard of care (e.g., widespread corticosteroid use). Furthermore, while the calibration was globally robust (as indicated by α and β), the remaining local lack of fit suggests the need for local intercept adjustment.

Among strengths, our study provides essential validation data from a Latin American (Peruvian) cohort, a region historically underrepresented in prognostic literature. Our study was conducted during a critical stage of the pandemic, in a context of high healthcare demand and severe clinical cases, and focuses on immunologically naive patients, providing a valuable baseline for response strategies against future novel pathogens or emerging waves. The finding that the simpler

q-CSI score demonstrated superior discrimination and feasibility -confirmed by the high data completeness reported in S1 Table- is highly relevant for triage and resource optimization in environments with limited access to complex laboratory variables. The methodological rigor, including the 100% double data entry and cross-verification protocol, substantially enhances confidence in our final dataset.

## Conclusions

In our resource-limited Peruvian cohort of unvaccinated patients hospitalized with COVID-19 pneumonia, the q-CSI demonstrated very good discriminative performance for 30-day mortality, surpassing the ISARIC-4C and SEIMC scores. The q-CSI stands out as a highly feasible and accurate tool for rapid triage, particularly when compared to the more wide-spread accepted ISARIC-4C, being the latter usually limited by data completeness in crisis settings.

## Supporting information

**S1 Table. Feasibility and data completeness across the four clinical prognostic scores (n = 1,963).** This table details the availability of each variable required for score calculation and identifies the primary laboratory or clinical parameters limiting the analysis for each model.
(DOC)

**S2 Table. Comparison of baseline characteristics between included and excluded patients.** This table compares the final analytic cohort (n = 1,074) with patients excluded due to missing data (n = 889) to assess potential selection bias across demographic and clinical variables.
(DOC)

**S1 File. Data set used to replicate study findings.** This dataset contains the clinical and demographic variables for the full study cohort (n = 1,963). It includes all parameters required to calculate the q-CSI, ISARIC-4C, SEIMC, and CALL scores, as well as the 30-day mortality outcome. Missing values are maintained to ensure transparency regarding the data structure and the selection process for the complete-case analysis.
(DTA)

**S2 File. Stata do-file for validation analysis.** This file contains the Stata code used to calculate the four mortality prediction scores, assess discrimination (AUROC comparison), and estimate calibration metrics (Hosmer-Lemeshow test, calibration intercept α, and calibration slope β) in the complete case cohort.
(DO)

## Author contributions

**Conceptualization:** Johan Azañero-Haro, Alonso Soto.

**Data curation:** Johan Azañero-Haro, Alonso Soto.

**Formal analysis:** Johan Azañero-Haro, Alonso Soto.

**Funding acquisition:** Johan Azañero-Haro, Alonso Soto.

**Investigation:** Johan Azañero-Haro, Alonso Soto.

**Methodology:** Johan Azañero-Haro, Alonso Soto.

**Project administration:** Johan Azañero-Haro, Alonso Soto.

**Resources:** Johan Azañero-Haro, Alonso Soto.

**Software:** Johan Azañero-Haro, Alonso Soto.

**Supervision:** Johan Azañero-Haro, Alonso Soto.

**Validation:** Johan Azañero-Haro, Alonso Soto.

**Visualization:** Johan Azañero-Haro, Alonso Soto.

**Writing – original draft:** Johan Azañero-Haro, Alonso Soto.

**Writing – review & editing:** Johan Azañero-Haro, Alonso Soto.

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
