## [Decision Letter · Decision Letter 0]

26 Oct 2025

PONE-D-25-29570Comparison of the performance of four clinical prediction rules for mortality in patients with COVID-19PLOS ONE

Dear Dr. Azañero-Haro,

Thank you for submitting your manuscript to PLOS ONE. After careful consideration, we feel that it has merit but does not fully meet PLOS ONE’s publication criteria as it currently stands. Therefore, we invite you to submit a revised version of the manuscript that addresses the points raised during the review process.

We look forward to receiving your revised manuscript.

Kind regards,

Frederick K Wangai, MBChB, Mmed (Int Med), FCP (ECSA), DHP

Academic Editor

PLOS ONE

Journal Requirements:

2. We note that your Data Availability Statement is currently as follows: “All relevant data are within the manuscript and its Supporting Information files.”

4. We notice that your supplementary figures are uploaded with the file type 'Figure'. Please amend the file type to 'Supporting Information'. Please ensure that each Supporting Information file has a legend listed in the manuscript after the references list.

Reviewer's Responses to Questions

**Comments to the Author**

1. Is the manuscript technically sound, and do the data support the conclusions?

Reviewer #1: Yes

Reviewer #2: Yes

2. Has the statistical analysis been performed appropriately and rigorously? 

Reviewer #1: Yes

Reviewer #2: Yes

3. Have the authors made all data underlying the findings in their manuscript fully available?

Reviewer #1: Yes

Reviewer #2: Yes

4. Is the manuscript presented in an intelligible fashion and written in standard English?

Reviewer #1: Yes

Reviewer #2: Yes

5. Review Comments to the Author

Reviewer #1: This is an excellent manuscript and thou the statistics at first glance appear complicated it simply and clearly explained. Well done.

Just to be sure,

Line 84: should it read referral or reference ?

Reviewer #2: Overview

This manuscript presents a retrospective cohort study comparing the predictive performance of four well-known COVID‑19 mortality prediction rules—q‑CSI, ISARIC‑4C, SEIMC, and CALL—in a Peruvian hospital cohort. The study design is clear and ethically sound, deriving a dataset from a larger previously published study. The authors focus on unvaccinated adults hospitalized with COVID‑19 pneumonia between March and December 2020 at Hospital Nacional Hipólito Unanue in Lima, Peru. By comparing the discriminatory and calibration performance of these scores, the study aims to address whether simpler models like q‑CSI can perform as well as, or better than, more complex scoring systems (e.g., ISARIC‑4C).

The main finding is that q‑CSI demonstrated the highest discriminatory ability (AUROC 0.85) with a favorable balance of sensitivity (86.3%) and specificity (70.9%) at an optimal cutoff identified via Youden index. ISARIC‑4C followed closely (AUROC 0.81; sensitivity 78.7%, specificity 69.1%) and showed the only acceptable calibration (Hosmer–Lemeshow p = 0.45). SEIMC and CALL had lower AUROCs and calibration performance, though each provided specific strengths—SEIMC displayed the highest specificity (76%), and CALL retained fair negative predictive value. The authors emphasize the need for simple risk stratification tools applicable in low-resource settings and discuss how their findings might inform clinical decision-making.

Strengths

Relevance and Timeliness: Assessing mortality prediction rules remains relevant because effective triage tools can optimize resource allocation during pandemic surges, especially in low- and middle-income countries. Although most global populations are now vaccinated, knowledge of score performance in unvaccinated contexts can inform responses where vaccination remains suboptimal. The focus on a Latin American cohort addresses a geographical gap in COVID‑19 prognostic research and provides data from a region heavily affected by the pandemic.

Clear Design and Ethics: The study’s retrospective design is well described and ethical approval is documented. The authors provide inclusion/exclusion criteria and sample size calculations (assuming 10% differences in sensitivity/specificity, 95% confidence level, and 80% power), then include all cases with complete data. Anonymization processes are noted. This transparency supports replicability and ethical compliance.

Comprehensive Statistical Approach: The statistical analyses are thorough. Variables were summarized appropriately based on distribution (means with SD or medians with IQR) and compared using parametric or non-parametric tests. The authors evaluate performance metrics (sensitivity, specificity, positive and negative predictive values, likelihood ratios) and discriminative ability (AUROC) with confidence intervals. They derive optimal cutoffs using the Youden index and assess calibration using decile-based plots, Spearman correlation, and the Hosmer–Lemeshow test. Pairwise comparisons of AUROCs are conducted to highlight differences relative to ISARIC‑4C.

Interpretation and Contextualization: The Discussion section contextualizes the findings by comparing them with prior validation studies across different populations. The authors note variability in AUROCs across regions and attribute differences to baseline cohort characteristics, therapeutic protocols, SARS-CoV‑2 variants, and the inherent tendency of models to perform better in their derivation cohorts. They rightly caution that calibration limitations can reduce clinical applicability even when discrimination is acceptable.

Limitations and Concerns

1. Selection Bias and Missing Data Handling: Only 1,074 of 3,074 patients (≈35%) had complete data for calculation of all four scores. Excluding two-thirds of patients risks selection bias. Missing data may not be random, and patients with incomplete records could differ systematically (e.g., severity, comorbidities, outcomes). The authors should compare baseline characteristics of included versus excluded patients and discuss how these differences might bias estimates. It would also be useful to explain why the sample sizes for each score (n=1844 for q‑CSI, 1408 for ISARIC‑4C, etc.) are larger than the final sample of 1,074—likely because each score had different missing variables. Clarifying whether multiple imputations or other strategies were considered would strengthen validity.

2. Generalizability: The cohort is single-center and exclusively unvaccinated. Given vaccination and new variants significantly change disease presentation and outcomes, the findings may not apply to contemporary COVID‑19 patients. The authors acknowledge this limitation but should expand discussion of how vaccination status, variant virulence, and changing treatment protocols affect predictive scores. Additionally, Peru’s healthcare system and patient demographics may differ from other Latin American and global contexts, limiting extrapolation.

3. Retrospective Data Quality: Retrospective chart reviews risk misclassification and documentation errors. The authors mention manual review of physical charts and data transfer to Excel, but further details about data quality control, inter-rater reliability, and training of abstractors would enhance confidence in the dataset.

4. Calibration and Model Updating: Although discrimination is the focus, calibration is crucial for clinical use. The q‑CSI, SEIMC, and CALL scores demonstrated poor Hosmer–Lemeshow fit, indicating risk predictions deviate from observed probabilities. The authors might explore recalibration or model updating tailored to their cohort. For example, logistic recalibration or refitting intercepts and slopes could improve predictive accuracy (Van Calster et al. 2019). If recalibration is out of scope, at least provide calibration intercepts and slopes or net reclassification improvement measures; these metrics are more informative than Hosmer–Lemeshow, which is sensitive to sample size (Riley et al. 2019).

5. Clinical Utility and Cutoffs: The decision to derive new binary cutoffs using the Youden index may optimize sensitivity and specificity but might oversimplify ordinal risk categories originally proposed for the scores. The authors should justify dichotomizing continuous/ordinal scores when original tools defined multiple risk strata for triage. Also, discuss how these new cutoffs would perform in alternative settings (e.g., outpatient triage) or under varying resource constraints.

6. Confounding and Unmeasured Variables: Mortality risk is influenced by many factors beyond those captured by scoring systems (e.g., time from symptom onset to admission, treatment availability, socioeconomic status, viral variants). Because the dataset originates from an earlier pandemic wave, factors like corticosteroid use or antiviral therapy may differ. The authors might discuss whether treatments or changes in standard of care during the study period confounded associations.

7. Sample Size Calculation and Power: The sample size calculation (582 patients) assumes 10% differences in sensitivity and specificity, but the reasoning could be better clarified. Since final included cases exceeded this threshold (1,074), power to detect smaller differences is likely adequate. However, specifying how missing data patterns affect effective sample size would help.

8. Language and Clarity: While the manuscript is generally well written, there are minor grammatical errors and awkward phrases. Examples include “The outcome primary was 30‑day in‑hospital mortality” (should be “The primary outcome was 30‑day...”), and “determine” is misspelled as “determinate”. The authors should proofread to improve readability. A professional language edit is recommended before publication.

9. Data Availability: The supporting information or link to a repository is not provided in the PDF. For transparency, deposit the dataset and statistical code in a public repository (e.g., Dryad, OSF) and include a DOI. If ethical or legal restrictions apply (due to patient privacy), provide contact details for an institutional data access committee.

Novelty and Contribution: Several studies have compared COVID‑19 prognostic scores; some recent meta-analyses have reviewed dozens of models (Wynants et al. 2020). The novelty here lies in directly comparing q‑CSI, ISARIC‑4C, SEIMC, and CALL in a Peruvian context. Although the unvaccinated single-center cohort limits broad applicability, the study adds local data and highlights that simple, respiratory-based scores may perform well in resource-limited settings.

Suggestions for Improvement

i) Describe excluded patients and missing data: Provide a table comparing demographics and outcomes of included and excluded cases to assess selection bias.

ii) Explain each score’s required variables and missingness: The differing sample sizes for each score (1844 for q‑CSI vs. 1408 for ISARIC‑4C etc.) suggest variable availability issues. Clarify which variables were missing and why.

iii) Consider model updating: Even a simple recalibration of intercept and slope could improve predictive accuracy; reporting these could encourage others to adopt similar updates.

iv) Expand on vaccination and variant context: Provide rationale for why unvaccinated data remain useful and discuss how the models might perform in vaccinated populations (perhaps referencing external validation studies).

v) Detail data quality control: Outline the training of data abstractors, double data entry or cross‑checks, and steps taken to minimize misclassification.

vi) Provide open data: Share anonymized data and code to comply with PLOS data policies and to allow other researchers to replicate or extend the analysis.

vii) Professional editing: Engage a native English editor to polish the manuscript and fix typographical errors.

REFERENCES

Riley, Richard D., Ewout W. Steyerberg, and Douglas G. Altman. 2019. “Better Reporting of Analyses Assessing Model Performance: Calibration Survival.” BMJ 365: l1821. https://doi.org/10.1136/bmj.l1821

Van Calster, Ben, Ewout W. Steyerberg, Maarten van Smeden, Laure Wynants, and Richard D. Riley. 2019. “Calibration: The Achilles Heel of Predictive Analytics.” BMC Medicine 17 (1): 230. https://doi.org/10.1186/s12916-019-1466-7

Wynants, Laure, et al. 2020. “Prediction Models for Diagnosis and Prognosis of Covid‑19: Systematic Review and Critical Appraisal.” BMJ 369: m1328. https://doi.org/10.1136/bmj.m1328

6. PLOS authors have the option to publish the peer review history of their article (what does this mean?). If published, this will include your full peer review and any attached files.

Reviewer #1: **Yes:** Shastra Avendra Bhoora

Reviewer #2: **Yes:** Miquel Angel Rodríguez-Arias

---

## [Author Response · Author response to Decision Letter 1]

14 Jan 2026

A rebuttal letter

Dear Frederick K Wangai, Academic Editor, and anonymous reviewers,

Thank you sincerely for your time, the thorough reading of our manuscript (PONE-D-25-29570), and the constructive and insightful comments you have provided. We believe that your suggestions and critiques have significantly strengthened the clarity and scientific rigor of our work.

We have carefully reviewed every point raised by the Academic Editor and the Reviewers, and we have made all necessary modifications in the revised manuscript.

In the following "Response to Reviewers Letter," we address each observation point-by-point. To facilitate the review process, we have copied and pasted the original comments in bold and italics and have presented our response immediately afterward, indicating the changes made in the manuscript (including the page or line number where applicable).

We hope that the revisions fulfill the high standards of PLOS ONE, and we thank you in advance for your consideration regarding the publication of our work.

Sincerely,

Reviewer #1

Reviewer's observation #1. This is an excellent manuscript and thou the statistics at first glance appear complicated it simply and clearly explained. Well done.

Just to be sure,

Response: We are highly gratified that the statistical methodology, while potentially appearing complex at first glance, was perceived as “simply and clearly explained.” We sincerely thank the reviewer for this positive assessment.

Reviewer's observation #2. Line 84: should it read referral or reference ?

Response: We appreciate this important point regarding terminology. We have reviewed Line 84 of the original manuscript and confirm that, in the context of patient transfer or moving a patient between different levels of care, the correct term is “referral”. We have corrected the wording in the manuscript to ensure precision and clarity.

Reviewer #2

1. Selection Bias and Missing Data Handling: Only 1,074 of 3,074 patients (≈35%) had complete data for calculation of all four scores. Excluding two-thirds of patients risks selection bias. Missing data may not be random, and patients with incomplete records could differ systematically (e.g., severity, comorbidities, outcomes). The authors should compare baseline characteristics of included versus excluded patients and discuss how these differences might bias estimates. It would also be useful to explain why the sample sizes for each score (n=1844 for q CSI, 1408 for ISARIC 4C, etc.) are larger than the final sample of 1,074-likely because each score had different missing variables. Clarifying whether multiple imputations or other strategies were considered would strengthen validity.

Response:

We sincerely appreciate your detailed and highly pertinent observation regarding the potential selection bias arising from the high proportion of missing data in our retrospective cohort.

1. We fully concur that the exclusion of a significant portion of the original patient cohort represents a limitation and introduces risk of selection bias. This high rate of data incompleteness is a direct reflection of the extreme resource constraints and clinical burden faced by our hospital, and the Peruvian health system overall, during the first wave of the pandemic. The resulting compromise in the quality of clinical documentation retrospectively underscores the pressing need for improved data capture systems in low-resource settings -a limitation that we feel merits explicit discussion in the manuscript, as it represents a negative impact on the quality of healthcare records.

2. Although 3074 patients were hospitalized, 2377 satisfied our general inclusion criteria. Among them, 1074 had compete information for calculating all four scores under evaluation (see flowchart). While we recognize the value of your suggestion to compare the baseline characteristics (demographics, comorbidities, and mortality outcome) between the patient with (n=1,074) and without complete information (n=1,303), we must regretfully report that this analysis is not methodologically feasible. The exclusion process occurred early during the data abstraction phase: patients whose charts were missing the minimum variables required for any of the four prediction rules were immediately filtered out. Consequently, their full baseline clinical characteristics and, critically, their mortality outcomes, were not extracted or digitized, precluding any comparison.

3. As the reviewer pointed out, the sample sizes for each individual score are larger than the final sample of 1,074 because each score had different missing variables.

To provide clarity on the distinct sample sizes (n) reported:

• n=1,074 (Primary cohort): This is the core comparative cohort. It is defined strictly by patients who had complete data necessary to calculate ALL four prediction scores concurrently (q-CSI, ISARIC-4C, SEIMC, and CALL). This group ensures a robust head-to-head comparison of all models.

• Larger “n” for Individual Scores: The larger sample sizes cited for individual scores (e.g., n=1,844 for q-CSI) reflect the differential availability of the specific variables required for each model. The q-CSI, requiring fewer and more common clinical and basic laboratory parameters, was applicable to a significantly larger subset of the initial pool compared to the ISARIC-4C, which relies on more complex inflammatory markers often missing from retrospective charts. This demonstrates that simpler scores are inherently more robust to data incompleteness in our setting.

4. Regarding alternative strategies, we also considered methods such as multiple imputation. However, given that the percentage of excluded patients (1,303 out of 2,377) represents 55% of the eligible cohort, we determined that multiple imputation would not yield statistically robust estimates. We therefore opted for the complete-case analysis, explicitly acknowledging and discussing its inherent limitations in the Discussion section.

Actions in the Manuscript:

- We will integrate these essential clarifications into the revised methods and discussion sections of the manuscript.

2. The cohort is single-center and exclusively unvaccinated. Given vaccination and new variants significantly change disease presentation and outcomes, the findings may not apply to contemporary COVID 19 patients. The authors acknowledge this limitation but should expand discussion of how vaccination status, variant virulence, and changing treatment protocols affect predictive scores. Additionally, Peru’s healthcare system and patient demographics may differ from other Latin American and global contexts, limiting extrapolation.

Response:

We thank the reviewer for raising this concern. We acknowledge that the generalizability of our findings is a key limitation due to the study's specific temporal and geographical context.

1. We accept that the single-center nature of our cohort, coupled with the fact that the study was conducted exclusively in unvaccinated patients during the initial waves of the pandemic (March to December 2020), inherently restricts the direct applicability of our results to current clinical practice. It is well-established that the development of population immunity, the emergence of new viral variants, and the standardization of therapeutics (such as corticosteroids and antivirals) have substantially altered the natural history, clinical presentation, and mortality risk of COVID-19. This evolution may indeed affect the predictive performance of scores developed or validated in earlier phases. We have expanded upon this critical point in the Discussion section.

2. Nevertheless, we believe the value of this study remains significant for several reasons:

• This validation study focuses on a Latin American (Peruvian) cohort, a region historically underrepresented in prognostic score validation literature. This provides essential data on the performance of these tools within a specific demographic and socioeconomic context.

• Our finding that simpler scores (like q-CSI) maintained high performance is crucial for triage and decision-making in low-resource environments, where access to complex laboratory variables necessary for scores like ISARIC-4C is often a significant logistical bottleneck.

• Data from unvaccinated or immunologically naive populations hold paramount importance for planning and response strategies against future public health emergencies that may involve novel pathogens or populations lacking prior immunity.

3. Finally, we will re-emphasize that the clinical application of these scores must always be understood within the context of hospitalized patient risk stratification. By design, their extrapolation to the general population (ambulatory patients or community triage) is, and remains, limited.

Actions in the Manuscript:

• The Discussion section will be expanded to incorporate a more detailed analysis of the impact of vaccination, variants, and evolving treatments on the predictive capacity of the scores.

• The justification for the study will be reinforced, highlighting the geographical gap and the utility for resource-limited settings.

3. Retrospective Data Quality: Retrospective chart reviews risk misclassification and documentation errors. The authors mention manual review of physical charts and data transfer to Excel, but further details about data quality control, inter-rater reliability, and training of abstractors would enhance confidence in the dataset.

Response:

We agree that data quality is an essential component of methodological rigor and that retrospective review, particularly in a public health emergency context, carries an inherent risk of transcription errors or misclassification.

To ensure the quality and reliability of our dataset, we implemented a comprehensive control protocol:

1. Data abstraction was carried out by the same core team of investigators who were simultaneously working as treating physicians in the hospitalization areas designated for COVID-19 patients. Their extensive clinical experience and direct knowledge of the clinical context were crucial for the correct and consistent interpretation of clinical and laboratory variables (including those obtained from handwritten records).

2. For rigorous transcription quality control, 100% of the abstracted data was subjected to a double data entry process (double digitization). Two team members independently entered the information into separate electronic files. Subsequently, a systematic cross-verification was performed to detect all discrepancies. Any differences identified between the two entries were immediately reviewed by a senior investigator and resolved by referring to the original patient record. This process resulted in a single, final validated dataset.

3. While we acknowledge that a formal measurement of inter-rater reliability was not performed, we consider this limitation to be substantially mitigated. Our confidence in the data quality is based on the unique combination of the deep clinical experience and contextual knowledge held by the physician/investigator abstractors, coupled with the methodological rigor of the systematic double data entry and cross-verification process.

Actions in the Manuscript:

• These specific details concerning the composition of the investigation team (treating physicians/investigators), double data entry, and the cross-verification protocol will be incorporated into the "Methods: Data Collection" section to enhance methodological transparency.

4. Calibration and Model Updating: Although discrimination is the focus, calibration is crucial for clinical use. The q CSI, SEIMC, and CALL scores demonstrated poor Hosmer–Lemeshow fit, indicating risk predictions deviate from observed probabilities. The authors might explore recalibration or model updating tailored to their cohort. For example, logistic recalibration or refitting intercepts and slopes could improve predictive accuracy (Van Calster et al. 2019). If recalibration is out of scope, at least provide calibration intercepts and slopes or net reclassification improvement measures; these metrics are more informative than Hosmer–Lemeshow, which is sensitive to sample size (Riley et al. 2019).

Response:

We thank the Reviewer for highlighting the importance of calibration and for guiding us towards the use of more informative metrics. We agree that the Hosmer-Lemeshow (HL) test is highly sensitive to large sample sizes, potentially masking adequate calibration performance.

We have addressed this point comprehensively by calculating and reporting the Calibration Intercept (alpha) and the Calibration Slope (beta) for all four scores in the complete case cohort (n=1,074).

1. Our analysis using alpha and beta demonstrates that the global calibration of all four models was robust, in contrast with the conclusion of 'poor fit' derived solely from the HL test.

• Calibration Slope (beta): The slope for all four scores was found to be statistically indistinguishable from the ideal value of 1.0. This demonstrates that the risk scale is structurally correct in our cohort.

• Calibration Intercept (alpha): The alpha values for the SEIMC (alpha=-0.04) and CALL (alpha=-0.03) scores were virtually zero. While the q-CSI (alpha=-0.14) and ISARIC (alpha=-0.13) showed a slight negative trend (suggesting minor risk overestimation), this deviation was not statistically significant (p > 0.05 in all cases).

We have expanded the Discussion to emphasize that these robust metrics (Table 2) confirm the models are globally well-calibrated, validating their use for risk stratification despite the negative HL result.

2. We acknowledge the importance of recalibration (e.g., Logistic Recalibration, as suggested by Van Calster et al.) for maximizing local utility.

• Since our primary objective was the direct external validation and comparison of the published models’ original performance, model updating or recalibration falls outside the scope of this initial study. However, we have made this explicit recommendation for future research based on the minor local deviations observed in the calibration plots.

3. Regarding the suggested to include the references by Van Calster et al. 2019 and Riley et al. 2019.

We believe the Van Calster et al. reference focuses on the methodology of recalibration and model updating, which falls outside the scope of our validation study. We regret to inform that despite efforts to locate the full text, we were unable to gain access to the work by Riley et al. (2019) to integrate its methodology. Therefore, we will instead cite the principles of calibration assessment based on available methodological literature (Stevens RJ et al 2020).

Actions in the Manuscript:

• The global performance table (table 2) in the result section now includes the Calibration Intercept (alpha) and Calibration Slope (beta) with their 95% CIs for all four scores based on the complete case analysis (N=1,074).

• The Discussion has been revised to argue that the alpha approx. 0 and beta approx. 1 findings indicate a robust global calibration, thus supporting the clinical application of these scores.

5. Clinical Utility and Cutoffs: The decision to derive new binary cutoffs using the Youden index may optimize sensitivity and specificity but might oversimplify ordinal risk categories originally proposed for the scores. The authors should justify dichotomizing continuous/ordinal scores when original tools defined multiple risk strata for triage. Also, discuss how these new cutoffs would perform in alternative settings (e.g., outpatient triage) or under varying resource constraints.

Response:

We appreciate this observation, as it addresses the crucial transition from statistical findings to practical clinical utility.

1. We recognize that the original scores define multiple risk strata, but our primary objective was to perform a direct (head-to-head) and standardized comparison of the predictive capacity of the four tools for the primary binary outcome (mortality). The use of the Youden index allowed us to establish an optimal and uniform cutoff point for each score, maximizing discrimination between "high risk of mortality" and "low risk of mortality" specifically within our cohort.

2

---

## [Editor Report · Decision Letter 1]

17 Mar 2026

PONE-D-25-29570R1

Comparison of the performance of four clinical prediction rules for mortality in patients with COVID-19

PLOS One

Dear Dr. Azañero-Haro,

Thank you for submitting your manuscript to PLOS ONE. After careful consideration, we feel that it has merit but does not fully meet PLOS ONE’s publication criteria as it currently stands. Therefore, we invite you to submit a revised version of the manuscript that addresses the points raised during the review process.

Essential revisions

**Missing data and selection bias (transparency)**

Please state explicitly in the manuscript what proportion of the source cohort was excluded due to incomplete documentation and clarify the implications for potential selection bias.If excluded records were not digitized/extracted and therefore included vs excluded comparisons could not be performed, please state this plainly so readers understand that selection bias cannot be quantified.Provide a brief missingness summary (preferably a supplementary table): proportion missing for key predictors used by each score and the resulting analytic sample size per score; include a simple participant flow (figure or table).

**Cohort definitions and “head-to-head” framing**

Several performance metrics are presented on different denominators across models, while AUROC/calibration are reported in the complete-case cohort. This is methodologically defensible but risks reader confusion.Please clearly distinguish: (a) “maximal available subcohorts per score” (not head-to-head) versus (b) the complete-case cohort used for direct head-to-head comparison, and relabel tables/Results text accordingly.

**Correct interpretation of sensitivity/specificity**

Please review and correct the Discussion text to ensure sensitivity and specificity are interpreted consistently with the outcome coding (e.g., if mortality is the positive outcome, sensitivity relates to correctly identifying deaths). Confirm explicitly which class is treated as the positive outcome.

**Endorsement claim**

The manuscript states that a score is supported/recommended by the World Health Organization. This is a strong claim and must be supported by a precise WHO source, or rephrased to a defensible non-endorsement statement (e.g., “widely validated and commonly used”).

Editorial/production requirements

Please remove all residual drafting artifacts (duplicated words, broken phrases, typographical errors) and correct any malformed in-text citation numbering. The next version should be a clean, publication-ready file.

Data availability statement

Please ensure the Data Availability Statement is internally consistent with what is actually provided (e.g., if a minimal dataset and code are included as Supporting Information, state this clearly and remove conflicting “upon acceptance” language unless required by journal policy).

Once these points are addressed, I anticipate the manuscript can be accepted without further external review.

We look forward to receiving your revised manuscript.

Kind regards,

Frederick K Wangai, MBChB, Mmed (Int Med), FCP (ECSA), DHP

Academic Editor

PLOS One
---

## [Author Response · Author response to Decision Letter 2]

4 Apr 2026

Dear Frederick K. Wangai, Academic Editor, PLOS ONE

Thank you for the opportunity to submit this minor revision. We appreciate your positive feedback regarding the strengthened data quality and calibration reporting. We have addressed each of the essential and editorial requirements point-by-point as follows:

1. Essential revisions

• Missing data and selection bias (transparency):

o Response: We have explicitly stated in the Results and Methods that out of a source cohort of 1,963 patients, 1,074 were included in the head-to-head analysis.

o Action: To address potential selection bias, we have added S2 Table, which compares the baseline characteristics of included (n=1,074) vs. excluded (n=889) patients.

o Clarification on Bias: The analysis in S2 Table revealed some statistically significant differences: the included cohort was slightly older (58.2 vs 56.4 years, p=0.011) and a higher prevalence of diabetes (27.9% vs 22.7%, p=0.008). We have acknowledged these differences in the Limitations section, noting that our analytic sample represents a slightly higher-risk group. However, we emphasize that since all four scores were evaluated on the exact same n=1,074 patients, the relative performance and ranking of the scores (head-to-head comparison) remain internally valid and robust for this population.

o Action: We provided a missingness summary in S1 Table, detailing the proportion missing for key predictors (e.g., CRP for ISARIC-4C score and LDH for CALL score) across the scores, which justifies the final analytic sample size and underscores the feasibility of simpler tools like q-CSI.

• Cohort definitions and “head-to-head” framing:

o Response: We have clarified the distinction between the "maximal available subcohorts" (used for feasibility) and the "complete-case cohort" (used for head-to-head comparison).

o Action: Tables and Results text have been relabeled to ensure clarity. We now clearly distinguish when referring to the total available data for a single score (e.g., n=1,844 for q-CSI) versus the core comparative cohort (n=1,074) used for AUROC and calibration metrics.

• Correct interpretation of sensitivity/specificity:

o Response: We have explicitly stated in the Methods section and in the footnotes of the results text that "30-day mortality" is treated as the positive outcome.

o Action: The Discussion (Interpretation of cutoffs) has been revised to ensure sensitivity is correctly interpreted as the ability to identify patients who will die (screening/Rule-out), and specificity as the ability to identify those who will survive (confirmatory/Rule-in).

• Endorsement claim:

o Response: We have removed the claim that any score is "recommended by the WHO" to avoid non-endorsement issues.

o Action: In the Abstract and Discussion, we rephrased this to: "a widely validated and internationally recognized reference standard often utilized in global clinical guidelines."

2. Editorial/production requirements

• Drafting artifacts and typos:

o Action: A complete manual review of the manuscript was performed. We corrected typographical errors such as “wich” to “which” and ensured all in-text citation numbering is sequential and correctly formed. The revised file is now a clean, publication-ready version.

3. Data availability statement

• Consistency:

o Action: We have updated the Data Availability Statement. S1 File now contains the complete dataset of 1,963 patients, including those with missing values, to allow for full reproducibility of our flow-chart and missingness analysis.

---

## [Editor Report · Decision Letter 2]

21 Apr 2026

Comparison of the performance of four clinical prediction rules for mortality in patients with COVID-19

PONE-D-25-29570R2

Dear Dr. Azañero-Haro,

We’re pleased to inform you that your manuscript has been judged scientifically suitable for publication and will be formally accepted for publication once it meets all outstanding technical requirements.

Kind regards,

Frederick K Wangai, MBChB, Mmed (Int Med), FCP (ECSA), FRCP Edinburgh

Academic Editor

PLOS One

---

## [Editor Report · Acceptance letter]

PONE-D-25-29570R2

PLOS One

Dear Dr. Azañero-Haro,

I'm pleased to inform you that your manuscript has been deemed suitable for publication in PLOS One. Congratulations! Your manuscript is now being handed over to our production team.

Kind regards,

on behalf of

Dr. Frederick K Wangai

Academic Editor

PLOS One